# Simulation as a Training Method for Electricity Workers’ Safety

**DOI:** 10.3390/ijerph18041591

**Published:** 2021-02-08

**Authors:** Fabrizio Bracco, Michele Masini, Donald Glowinski, Tommaso Piccinno, Simon Schaerlaeken

**Affiliations:** 1Dipartimento di Scienze della Formazione (DISFOR), Department of Education Sciences, University of Genoa, 16126 Genova, Italy; masini@vie-srl.com (M.M.); piccinno@vie-srl.com (T.P.); 2Valorizzazione, Innovazione, Empowerment (VIE SRL), Spinoff of the University of Genoa, 16126 Genova, Italy; Donald.Glowinski@unige.ch; 3Neuroscience of Emotion and Affective Dynamics Lab (NEAD), Affiliated to the Swiss Center for Affective Sciences (SCAS), University of Geneva, 1205 Geneva, Switzerland; simon.schaerlaeken@gmail.com

**Keywords:** simulation, non-technical skills, industrial safety

## Abstract

Background: Simulation is a useful method to improve learning and increase the safety of work operations, both for technical and non-technical skills. However, the observation, assessment, and feedback about these skills is particularly complex, because the process needs expert observers, and the feedback could be judgmental and ineffective. Therefore, a structured process to develop effective simulation scenarios and tools for the observation and feedback about performance is crucial. To this aim, in the present research, we developed a training model for electricity distribution workers, based on high fidelity simulation. Methods: We designed simulation scenarios based on real cases, developed, and tested a set of observation and rating forms for the non-technical skills behavioral markers, and we tracked behaviors based on non-verbal cues (physiological and head orientation parameters). Results: The training methodology proved to be highly appreciated by the participants and effective in fostering reflexivity. An in-depth analysis of physiological indexes and behaviors compliant to safety procedures revealed that breath rate and heart rate patterns commonly related with mindful and relaxed states were correlated with compliant behaviors, and patterns typical of stress and anxiety were correlated with non-compliant behaviors. Conclusions: a new training method based on high fidelity simulation, addressing both technical and non-technical skills is now available for fostering self-reflection and safety for electricity distribution workers. Future research should assess the long-term effectiveness of high-fidelity simulation for electricity workers, and should investigate non-invasive and real-time methods for tracking physiological parameters.

## 1. Introduction

Simulation is one of the most acknowledged and widely used methods to train adults in many domains with the goal of promoting safety and performance efficacy on the workplace [1]. It is considered one of the best methods to transfer new knowledge, skills, procedures, and to shape behavior [2,3]. Among its advantages, we see its potential to foster reflective practices in workers. Simulation provides the opportunity for workers to engage in a situation which is similar to the usual work activity, but has a different purpose: not to have the job done, but to reflect on what has been done for the sake of learning and improving one’s professional skills and attitudes. Therefore, after a simulation, workers can think about their performance, what was safe, compliant, or reckless, and they slowly pass from “reflection-on-action” (thoughts about their performance), to “reflection-in-action” (i.e., the capacity to critically reflect on what is happening in the very moment it is happening) [4]. Reflective capacity has been considered of paramount importance also for improving safe performance, safety climate, and the capacity to perform safely adapting the procedures in dynamic environments [5,6]. Taking into account modern work activities, simulation has been introduced in the aviation domain since the beginning of the 20th century. Its effectiveness as a training method has been so well recognized that it progressively spread to other domains like healthcare [7], maritime transportation [8], road safety [9], industry [10], and construction [11].

According to McGaghie and colleagues, simulation has the potential to transfer in the current practices the technical and non-technical aspects learned during the simulated scenario [9]. Non-technical skills (NTS) are complementary to the technical skills that contribute to the activation of safe and effective work performance [12]. One of the most recognized categorizations of NTS divides them into two subsets: cognitive and social. The cognitive NTS are situation awareness, decision making, and task management, while the social NTS are communication, leadership, and teamwork [13,14]. They represent all the skills not pertaining to the technical expertise of a given profession, but that play a key role for both work quality and safety [14]. Notwithstanding several models have been proposed, we are still lacking a comprehensive model of all these skills necessary to guarantee the accomplishment of safe operations. This is probably due to the fact that the set of NTS widely vary according to the specific work environment and the specific activity [15]. 

Technical and non-technical skills are both exceptionally important for safe operations and the development of safety culture [16,17]. Skills can be improved through experiential training and structured reflection, and simulation is a valid method for this purpose. Simulation has been adopted as a training method in the electrical domain as well, even though it is mostly used as a training tool for new hires and apprentices, mainly focused on specific technical procedures. Simulation training for safety has been described in [18], in order to increase electrical hazard awareness of construction workers. They developed a virtual environment simulation training program with interactive tools aimed at engaging workers in safe work procedures. 

To the best of our knowledge, the training of skills that are not merely technical or procedural in the electricity domain is not quite widespread. Saurin seems to be the only researcher that published papers on scenario-based training of NTS for electricity operators. In [19], the authors conducted several interviews of electricians following the Critical Decisions Method, i.e., asking them to explicitly verbalize their decisions and thoughts in several critical conditions. In addition, they spent several hours on the field, observing the workers during their every-day activity. As a result, they outlined a list of 12 NTS for electricians. The list is presented below:
To discuss with the operation team defining procedures that should be taken and obtaining line information.To discuss with the field team reaching a common understanding on the situation.To discuss with the consumers and population the status and risks of line maintenance procedures.To express doubts, fears, and requests for help from other team members.To identify structure, lines or equipment that are non-standard, damaged or have failed.To identify visible signs in the environment that indicate difficulties in doing the task or the probable cause of damages to the line.To develop strategies to identify line defects.To develop work strategies, after the defects have been identified.To plan and to check the equipment and materials that are necessary to undertake the task.To distribute the tasks between team members and to do the task accordinglyTo identify causes of stress and fatigue.To develop strategies to cope with stress and fatigue situations.

In a further research [20], they proposed a framework for the design of scenario-based training of grid electricians within the resilience engineering perspective. The same framework has been further tested in a comparison between physical and virtual scenarios, demonstrating that operational contexts based on computer interfaces could be effectively transferred into virtual simulations [21].

Following this line of research, the aim of this study was therefore to develop and test a set of observation and rating forms for the technical and non-technical behavioral markers of workers involved in a simulation of electric tasks. In addition, we wanted to enrich the analysis of the performance not only through peer-observation, but also by means of the monitoring of behavioral and physiological parameters.

### 1.1. From NTS to Situated Professional Skills

Literature differentiates between technical and non-technical skills [22]; however, for the purpose of training and fostering a safety culture, dividing technical and non-technical skills could lead to the misunderstanding that technical skills are the core of the job, while non-technical skills are something relevant but non clearly applicable to the work activity, some rather vague concept upon which all agree but lacking a thorough training framework. The term “non-technical skills” has been criticized by Nestel et al. [23] because it refers to a “deficit model” where these skills are defined as a negative (or even opposite) side of technical skills. As the authors claim, the “goal is to use encompassing and positive terms to describe the complex sets of skills required for safe and competent clinical practice” [3,23].

In this research, we tried to overcome the dichotomy between technical and non-technical skills, and we framed professional competence as formed by three interconnected pillars, which cannot stand alone: (a) operative competence, which is technical skills and knowledge; (b) personal competence (both emotional and cognitive), which is made of abilities such as decision making, problem solving, situational awareness, workload and stress management; and (c) social competence, which consists in being able to communicate and work with others in an effective way. The least two aspects are usually referred to as NTS, but we decline this definition to underline the complementarity and indivisibility of these three aspects. 

Technical skills are abstract and decontextualized, and they are usually learned in formal educational and training processes. They must be adjusted to match real world conditions: this match is made by the use of personal and social skills. Any task of any job involves somehow all these three aspects of the professional competence. We define these abilities as Situated Professional Skills (SPS) because they represent the application of the skills of a practitioner in his own every-day work. This terminological proposal draws back to the theory of situated cognition, which implies that cognitive processes need to be understood and framed taking into account the specific cultural, physical, organizational, and operational situation [24]. SPS are usually learned with practice, on the job training, and with coaching. In this research, we aimed at developing a training framework to foster SPS in the electricity distribution domain by means of simulation. 

### 1.2. Simulation-Based Training for SPS in the Electricity Distribution Domain

Simulation has been extensively proved to be an effective method for fostering reflective practices about SPS and therefore reinforcing safe attitudes and behaviors [25,26]. 

Notwithstanding the evidence of the relevance of SPS for the safe and efficient accomplishment of operations, the observation, assessment and feedback about these skills during simulation is particularly complex, because the process needs expert observers, and the feedback is often provided in judgmental and ineffective ways [27]. 

However, the effectiveness of simulation is strongly dependent on the debriefing session, since learning actually occurs when it is based on proper feedback aimed at stimulating reflection about individual and team dynamics. The ability to analyze each other’s performance retrospectively is crucial when it is focused not only in talking about what went well and what did not, but also on why it went well and why something else did not [28,29].

Therefore, a proper tool for observing specific behaviors during simulation scenarios is of paramount importance, especially for the balanced training on technical and non-technical skills. Literature is rich of tools for the observation of teamwork and communication during simulated scenarios, but they are usually specific for a given professional domain (e.g., healthcare [30] aviation [31], railways [32], etc.). There is a lack of structured tools for the observation of SPS in electricians working on electricity distribution power lines. Moreover, at the best of our knowledge, NTS-oriented simulation has never been adopted in a technical activity such as electric distribution operations.

Taking into account the works of Saurin and colleagues [19,20,21], we decided to apply their method to the Italian electrical work context, and we developed a training framework based on simulation. The inclusion of behavioral and physiological parameters in high fidelity simulation is not very widespread and is mostly applied to the healthcare context. Some authors [33], reported significant correlation between performance in ultrasound simulator and cognitive load (measured with eye-based physiological indices, and behavioral measures). Others [34] correlated several stress indexes: self-reported (feeling stressed), biochemical (plasma cortisol), and physiological (heart rate and heart rate variability) during a simulated resuscitation and found that self-reported stress was the only predictor for low performance. Others [35], followed a similar approach and correlated performance in clinically based emergency scenarios with self-report and physiological stress indexes, but with the aim of establishing the effect of simulation-induced stress upon learning. The authors found that simulation-induced stress decreased when the simulation was repeated over time and learning consolidated.

In this research, we aimed at developing a training framework for electrical workers focused on safe and compliant behaviors during standard activities. In addition, we collected physiological and behavioral parameters for a pilot study about the correlation of these indexes with safe and compliant behavior. In summary, we wanted to: build a SPS training toolkit based on high-fidelity simulation;correlate the observation of SPS with data concerning physiological parameters.

## 2. Materials and Methods

### 2.1. The Definition of SPS for Electrical Workers

According to Thomas [14], the development of an assessment system should be based on evidence coming from some of the following sources: (i) review of the scientific literature, (ii) accident and incident analysis, (iii) cognitive task analysis, (iv) interviews/focus groups with expert operators. In our study, we mainly relied on literature review and participatory-design interviews with expert operators. An outline of the materials preparation is provided in Figure 1 below.

We first performed an extensive literature review about NTS and simulation training for electricity distribution operators (EDOs). Due to paucity of published papers on simulation of electricity distribution scenarios and the absence of research about evaluation of electricity distribution operators’ NTS, we based our study on the literature of NTS observation tools in healthcare [36] and the work of Wachs et al. [19].

We started our investigation on other professions (e.g., healthcare practitioners), taking into account the tools that were developed for simulation-based training. Taking into account the work by Wachs et al. [19], we adapted items referring to NTS for healthcare professionals (situational awareness, communication, teamwork, decision making, leadership) adapting them to the electricity distribution context. The behavioral markers obtained were therefore SPS indicators for EDOs, and they could be used as observational tools for simulated activities. 

We held a series of meetings with professionals that were belonging to different professional profiles. The professional profiles involved in the research (four engineers and two operators), thereby working on the adaptation of the tools provided in Wachs et al. [19] to the Italian electricity distribution context. Each meeting was held by three expert psychology researchers and involved at least one or two electricity distribution experts. 

Integrating the NTS with the technical aspects of the EDOs tasks, we developed a first draft of the tool based on a list of SPS such as: Knowledge of expected conditionsObservation of real conditionsUnderstanding real conditionsImplementation of safe working conditionsThird-Party communicationMaintaining attention despite disturbancesTeam communication/collaborationDocumentation usageStopping the work due to possible fraudStopping the work due to unsafe conditions

Then, we led two participatory-design meetings with a new sample of five practitioners to validate the content of the items and to check their comprehensibility. All the participants were male expert workers (>10 years of experience) with different roles (operator, team leader and engineer). During the first meeting a facilitator went through the items one by one and asked the participant to explain them. Participants were encouraged to discuss the meaning of the items and their relevance for safety. An observer took note of the relevant information. As a result, unclear items were rephrased. Furthermore, some items were added to encompass aspects that were not considered in the first draft. The second meeting was conducted in the same way with the same participants to confirm the refined version of the instrument was clear, easy to understand, without any ambiguity or inconsistency, and complete. Finally, we designed a tool for the overall assessment of the performance of electricity distribution operators. With the consensus of the experts, the items explored three main areas concerning the operators’ tasks: task management, customer management, and teamwork. 

The main goal was to list only the observable behavioral markers, while avoiding items that were too generic or that were inherent to unobservable mental states [37]. Since the list of items had to be suitable for the Italian context, we asked the practitioners to describe their tasks. We tried to develop a tool that was easy to use during the debriefing session of the simulation training aiming to make it quick to complete and easy to understand for practitioners who are unskilled in debriefings and NTS. At the end of the development process, we developed a list of 14 items divided into three domains, called SPS simulation Checklist. 

We defined detailed behavioral markers for each item, in order to anchor the performance rating on a three-point rating scale. Each level of performance (poor, medium, and good) had a specific behavioral marker defining the observable performance.

We adopted this kind of scale because, in a previous test, we observed that a dichotomous scale was perceived to be too judgmental by respondents, since they only had two options, i.e., good or poor performance. Therefore, we adopted a three-point scale, which presented a simple layout articulated enough to catch the information about the performance of the participants to the simulation training.

All the items underwent an iterative refinement process in order to reference only observable behaviors. For instance, instead of asking if every team member knew about his or her role (which is not observable), the item asked if the roles were clearly and explicitly identified (see Appendix A) reports the items and the behavioral markers). 

### 2.2. Development of Simulation Scenarios 

Simulation needs to respect some characteristics to reach its training goals [38]. Scenarios must be realistic in terms of script and simulation environment. Immersion has to be high. The term immersion, derived from virtual reality simulation, refers to the level to which the participants to the scenario lose their feeling of being involved in a simulation and perceive the events in the scenario as similar to the everyday experience [39]. The higher the immersion, the more likely the observed behavior will be similar to that displayed in normal work conditions. 

Even if some features of the simulation could not be the same as in the real world, there are some countermeasures to implement to keep immersion high. According to Hagiwara and colleagues [40] a proper scenario design should take into account the events and dynamics which could hinder immersion. For example, destructive interaction between participants and persons outside the scenario, or unnatural interaction with the client and/or another person in the scenario, or disturbance of technology that is not part of the simulated environment (e.g., a microphone).

We followed a backward design to develop the scenarios: we started defining the educational objective and then we went back until the definition of the script [41].

The methodology of the backward design consists of the following phases for each scenario:
Clearly define the educational objectives with experts. Objectives have to be adequate to the knowledge and skills of the participants and to the activities and tasks usually performed. We decided to focus on the most common medium-low voltage tasks.Identify which SPS are involved in each scenario and define the equivalent behavioral markers (BM). Before starting the writing of the script, we should be aware of which behavior indicates a good performance.Precisely define the script of the scenario in such a way that it requires the BM previously identified. Most of the time, scenarios are adaptation of real cases involving accidents or quasi-accidents (events when the accident did not happen but it could have easily happened). All the scenarios were analyzed by the experts in their professional experience. They are modified to explicitly require the BM related to the objectives of the scenario. Not realistic situations are avoided because they could generate not realistic behaviors.Development of the simulation room, using real tools and equipment modified, if needed, to reach the goal of the simulation. Realism is a priority.

Five scenarios have been developed (see Appendix A). Each scenario has been described according to this schema:Task AssignedGeneral Scenario DescriptionOverall Learning ObjectivesSPSOperational ContextMaterialsSetting Set-upParticipantsScenario Saving ItemsExpected DurationDetailed Sequential Description of Scenario

Taking into account the SPS and the observation checklist, the five scenarios were designed in order to cover a wide range of situations.

In addition, in order to help the trainers to observe the key points of participants’ behavior during the simulation, we designed for each scenario a detailed schedule of triggers, events and related SPS involved (Table 1). 

### 2.3. The Standardized Client

Relational SPS can be assessed through the relationship operators have with their colleagues, and with other roles that may be involved in the simulation, both indirectly (team leader, Central Operation Room), and directly (clients, passers-by, other professionals not working in the operator’s company). In both cases, the representation of these roles is transferred to the instructors, which should interact with the participants in order to trigger some specific behavior coherent with the learning objectives. Usually, the client’s role is to engage participants in a particular relationship (conflictual, unsafe, cooperative, etc.), which will be one of the main topics of reflection during the debriefing. 

The simulated client is a critical role [42], and his script must be carefully designed specifying aspects such as: learning outcomes it should aim for, specific behaviors during the scenario, etc. 

### 2.4. Development of Non-Verbal Cues (NVC) Tracking System

On the training ground, the participants were the focus of multiple sensors and recording devices. Our challenge was to develop a wearable multimodal setup which could be deployed fast (in less than one hour) in a variety of indoor professional locations. Both the video and the audio were recorded from different angles (see Figure 2). The audio was recorded from three different devices: from Xiaomi, the external microphone Zoom H6 for the surrounding soundscape, the action camera built-in microphone (YI 4K Xiaomi) as a back-up and finally directly on a microphone placed next to the subject’s mouth (Boya By-M1 lavalier microphone) to monitor each individual utterance distinctly. All the audio streams were sampled at 48 KHz. The video setup follows two different approaches. The action camera takes the subject’s perspective and shows what he is actively looking at since it is mounted directly on his helmet. Our connected cameras (CCTV-type, 140 field of view, 1080p resolution) are placed to give four different fixed perspectives on the scene giving a general view of what is happening.

Finally, we measure a number of physiological data using Zephyr belts placed directly on the skin of the torso. The physiological data of interest comprises heart rate, breathing rate, heart rate variability, and activity measures. The heart rate and breathing rate measures respectively the number of beats per minute and number of breaths per minute. Monitoring heart rate as well as breathing rate is usual practice nowadays in sports and intense activities [43]. It is usually used as an indicator of an individual’s training status and is impacted by the type of activity performed. The heart rate variability is a measure of the variation in the time interval between individual beat detections. This variability has been regarded as a proxy for “vertical integration” of the brain mechanisms that guide flexible control over behavior with peripheral physiology, and as such provides an important window into understanding stress and health [44]. Lastly, the activity is a direct translation of the movement of an accelerometer and symbolized by vector magnitude units showing how the various axis points are added (lateral, sagittal, and vertical); ~0.2 VMU for walking and ~0.8 for running. This last parameter can be used to explain part of the variance underlying the different measured rates (heart and breathing). 

The choice of using such belts and measuring these specific parameters falls behind three rationales: the ease of use, the visualization, and the literature associated. First, the belts are easy to set up, small, and portable. They work wirelessly and show impressive battery life. They are furthermore non-invasive, and allow the technicians to move freely. Second, the software provided with the belt is visually easy to read, with information displayed in real time. The evolution of the different measured rates is also displayed on screen for easy read out. Multiple belts can be used and will be displayed separately in the software window. Third, as mentioned, these parameters have been explored in multiple contexts of physical activities and stressful events. Two other parameters related to stress, electrodermal activity and pupil dilation, were left out of this experiment since they required additional devices that would clutter too much the non-invasiveness of this experimental setup. 

### 2.5. Debriefing Room 

The debriefing room allows for external viewers, members of the evaluation panel, and co-workers to evaluate the performance of the subject without being directly present on the training ground (Figure 3). All of this is supported because video recordings, audio recordings, and physiological data are streamed from the training ground to the debriefing room. Every signal recorded on the training ground is sent to an external computer (except the LIFE cameras that are connected to their own house-designed hub). The computer receives a live stream of the footage acquired by the Action Cam YI 4K by Xiaomi over a local WIFI signal. This live stream was supported by the mobile application Vysor. We also display a live update of the physiological data acquired at a frequency of 1 Hz. Both signals are subsequently rearranged over the space of one screen. Finally, this screen is duplicated via HDMI cable to a screen in the debriefing room. The recordings from the four LIFE cameras are mixed with the audio coming from the mouth microphone (amplified thanks to a sound amplifier) and are sent over HDMI cable to a beamer and multiple speakers in the debriefing room.

This configuration offers a complete overview of the action happening on the training ground with both overall and detailed representations of the operations at play. Based on such a rich environment, the expert panel and co-workers can give constructive feedback on the subject’s actions.

After the recording sessions, the different media were put together in “multi-views” videos. This created a single media that displayed the point of view from the four “life” CCTV-type cameras, the frontal action camera, as well as the evolution of the technicians’ biosignals. These multi-views videos were created using Adobe Premiere. Faces were also in some cases blurred to protect privacy. This material allows for a global understanding of the task at end and could be used as learning material for the technicians. A complete list of devices is available in Appendix A (Appendix A).

## 3. Procedure

The whole simulation session is managed by psychologists with expertise in high-fidelity simulation in cooperation with technical personnel of the electricity distribution company. After a preliminary introduction, the participants were informed about the goals of the research and signed an informed consent, according to the American Psychological Association ethical guidelines. After that, the participants visited the simulation room in order to familiarize with the environment. They were told how to operate the devices, which during the simulation they would have worked with the electricity turned on (respecting safety regulations), which they should have behaved “as if” they were in a real work situation. This phase was crucial to let the participants immerse themselves in the scenario, understanding the limits and characteristics of the simulation (e.g., the use of the work cell phone, the ICT tools, the task documentation and the devices). 

Every participant took an active part in at least one scenario and was an observer of all the others. The participants in the debriefing room could watch the simulation on a wide screen with multiple perspectives from the several cameras (Figure 4). In addition, the observers had a dedicated monitor for the physiological parameters. The timeline of the procedure is outlined in Figure 5 below. A complete checklist of tasks to be performed before and after the simulation is available at Appendix A. 

### 3.1. The Setting

The setting was the simulation center of the electricity distribution company that took part in the project, usually adopted as a training center for newly hired operators. The simulation room was equipped with electricity meters and other devices that are typically part of operators’ job. Special boards were used to separate the areas and reproduce small spaces (e.g., a basement where the meter is installed).

The cameras were positioned in order to have a full view of the scene from many perspectives, plus the action camera on operators’ helmets. The operators that took part in the scenarios also had a wearable microphone, in order to leave them free to move and reduce the threats to immersion. In Figure 6 are presented some pictures of the simulation center.

### 3.2. The Debriefing Phase

After the scenario the participants joined the rest of the team in the debriefing room. They were greeted with an applause, to make the climate warm and open to discussion. They sat in a circle and were facilitated in the debriefing by a psychologist expert in simulation. After the participants described their actions, the observers were invited to add suggestions and comments, facilitating a team reflection on the scenario. This whole phase was conducted using the SPS checklist, enabling the participants and the observers to anchor their comments to specific, observable behaviors. When necessary, the video recording of the simulation was played in the debriefing room, in order to reflect on what has been done, what was remembered, and why. The outline of the debriefing is presented in Appendix A (Appendix A).

### 3.3. Data Analysis Method

The five scenarios have been assessed with three methods:Training efficacy evaluationsubject matter expert (SME) assessment of the performance compliance to safety procedures through video analysisNVC assessment through sensors recording physiological data

The first assessment aimed at getting the feedback from participants right after the simulation training, in order to understand its usefulness in this work domain and its suitability for triggering self-reflective processes. The second and third methods, aimed at investigating a possible correlation between safe and compliant behaviors with physiological and behavioral parameters recorded during the simulation.

## 4. Results

### 4.1. Perceived Usefulness and Satisfaction of Participants

We adopted the Kirkpatrick’s Evaluation of Training Model [45] to assess the efficacy of the training method. Specifically, we focused on the reaction to the training, i.e., to what degree participants react favorably to training. An 11-item post-training feedback form was administered immediately after the end of each training session. The form aimed to evaluate the experience of the participants. The items measured three main areas: (1) quality of the training, (2) perception and feeling of the participants, and (3) perceived usefulness. The questions used in the form are available at Appendix A. 

Participants were asked to answer on a five-point Likert scale (Not at all, A few, Moderately, A lot, Completely).

A one sample t test was performed for each item to evaluate if the mean score was significantly different from the median point of the response scale (3). Table 2 shows all items had an average score statistically higher than 3. Participants considered the training as a valuable event: scenarios were perceived as realistic, participants felt engaged and satisfied and the training was evaluated as useful.

### 4.2. SME Assessment of Performance

In order to have a detailed analysis of the behavior of each participant to the simulation, we performed a further assessment with the support of an expert of the electrical operations, provided by the electricity distribution company. The assessment consisted in the analysis of each video recording, tracking each behavior and classifying it. The classification was based on the procedures of the company known by the expert. The expert evaluated if they were completely respected (compliant), if some minor issue were detected the behavior was labelled as partially-compliant, or if there were major violations it was labelled as non-compliant. By compliant we meant that the observed behavior was completely adherent to the operational rules and procedures for that specific activity. The behavior was rated as partially compliant when the actions were not completely adherent with the procedures, because some steps in the procedure had been missed or switched. The behavior was rated as non-compliant when the procedure was completely neglected, or the workers skipped some safety-critical steps in the procedure. 

The analysis of each scenario lasted between 4 and 5 h, and consisted in the observation step by step of the simulation, and deep discussion with the SME, facilitated by a psychologist. In Appendix A we reproduce an example of analysis; the complete set of analyses is available upon request.

### 4.3. NVC Physiological Data Analysis

The videos and NVC physiological data were divided into 15-s clips. For each of these clips, three levels of compliance to safety procedures (compliant, partially compliant, and non-compliant) were evaluated by subject matter experts (SME) through video analysis based on field expertise and years of practice. The NVC physiological data was also summarized at the clip level using multiple descriptive statistics: mean, median, maximum, minimum, and standard deviation. 

First, we computed correlations to measure how pairs of physiological data related to one another. Second, we used Linear mixed models (LMM) to analyze how the physiological NVC data differed across the various levels of compliance. LMMs use random effects modelling to improve the accuracy of the model. In our case, we used the participants (electricians in training) at random to take into account inter-individual differences. We used chi-square difference tests to investigate the contribution of the various levels of compliance to model the physiological data. Both analyses were performed with the R statistical software. 

In order to facilitate the comparison, we displayed the patterns of correlation between the NVC in the three conditions (compliant, partially compliant and non-compliant) (Figure 7a–c). In this section, we focused on the average value for each NVC measure (mean). In the compliant condition, heart rate was correlated with breathing rate (Pearson’s r = 0.36, *p* = 0.013) and with activity (*r* = 0.42, *p* < 0.001), and negatively correlated with heart rate variability (*r* = −0.89, *p* < 0.001). Breathing rate was also correlated with activity (*r* = 0.3, *p* < 0.001) and negatively correlated with heart rate variability (*r* = −0.29, *p* = 0.002). In the partially compliant condition, heart rate was not correlated with breathing rate (*r* = −0.03, *p* = 0.94) neither with activity (*r* = 0.05, *p* = 0.37) but still negatively correlated with heart rate variability (*r* = −0.68, *p* < 0.001). Breathing rate was however correlated with activity (*r* = 0.25, *p* < 0.001) and heart rate variability (*r* = 0.16, *p* = 0.04). In the non-compliant condition, heart rate was inversely correlated with breathing rate (*r* = −0.43, *p* = 0.008), not correlated with activity (*r* = −0.006, *p* = 0.26), and still negatively correlated with heart rate variability (*r* = −0.80, *p* < 0.001). Breathing rate was still correlated with activity (*r* = 0.32, *p* < 0.001) and heart rate variability (*r* = 0.56, *p* < 0.001).

Using LMM, we showed that using the levels of compliance as a fixed effect statistically improved the modelling of the average value of heart rate (χ2HR (5, N = 550) = 152.25, *p* < 0.001, Rm2 = 0.09), activity (χ2activity (5, N = 550) = 16.44, *p* < 0.001, Rm2 = 0.03), and heart rate variability (χ2activity (5, N = 550) = 105.66, *p* < 0.001, Rm2 = 0.10). It did not help modelling the breathing rate (χ2BR (5, N = 550) = 3.73, *p* = 0.15, Rm2 = 0.007). The LMM also highlighted significant differences between each of the three compliance conditions for heart rate and activity (Figure 8). The heart rate was significantly slower for the compliant condition compared to the partially and non-compliant conditions (*p* < 0.001) while activity was significantly lower for the non-compliant state compared to the compliant and partially compliant conditions (*p* < 0.01). Heart rate variability gradually decreased as the situation became less and less compliant with security measures (*p* < 0.001).

## 5. Comments

The data analysis concerned the post-training feedback form and the physiological data. The results of the feedback form show an overall satisfaction of the participants about the training. It was the first time they attended this kind of training, both for the method and the goal. Simulation was not previously used a recurrent training method and the learning objectives have never been focused on non-technical aspects of the tasks. Notwithstanding its novelty, the participants rated very positively this experience. As widely discussed in literature, one of the main risks of debriefing is the judgmental content of the feedbacks [27,28]. The participants evaluation of the non-judgmental approach of the course was extremely positive, confirming the effectiveness of the SPS Checklist as a tool for the correct observation of behaviors and proper feedback provision during the debriefing. Another critical aspect of the simulation is its realism and its capacity to favor immersion, helping the participants to feel like they were actually involved in a real work activity. A feeling of inauthenticity could hinder the effectiveness of the method, downgrading simulation as a “playful” and fake reconstruction of work, supporting the idea that what happens in a simulation is not what happens in reality. This would disrupt every reflective process, because the reflection on the simulated behavior would not create the ground for the reflection during real work activities. The participants rated the feeling of engagement in the situation, the realism of scenarios’ contents and the simulated environment as very positive, confirming that the situation effectively represented real work conditions. We also wanted to investigate the perceived usefulness of all the phases of the training session: introduction to the training, participation to the scenarios, observation of other colleagues, and debriefing. All these phases were rated very positively, especially the debriefing. This is a further evidence of the perceived usefulness of a structured moment in the training to guide reflective practices. Finally, to investigate the opportunity to implement simulation as a regular training method of the company, we asked the participants their intention to participate to further similar events and the overall satisfaction of this kind of training. The ratings were extremely positive, confirming the relevance of reflective practices for the sake of safety culture and professional development. 

Concerning the physiological data, the results show correlation patterns that are different according to the compliance to procedures. Distinct underlying physiological trends can thus be identified between compliant and non-compliant behaviors. The main potential advantage of this outcome is to use this information to help professionals raising their awareness about such covert processes and anticipate error-prone behaviors. We detail the results in the following.

### 5.1. Compliant Situations and Physiological Data

When the workers are compliant with security procedures, BR and HR are positively correlated and HR is comparatively lower than in the other non-compliant situations. This result is interesting, since it is a cue suggesting that the workers, in that moment, were in a particularly calm and mindful state of mind. A coherent BR, associated with a lower HR, seems to be associated with a psychological condition of mindfulness and well-being. As suggested by [44] in their systematic review of the literature about the psychological effects of slow breathing techniques, a decrease in breath rate can “enhance interactions between autonomic, cerebral and psychological flexibility, linking parasympathetic and Central nervous System activities related to both emotional control and well-being” [44] (p. 10).

These physiological conditions are commonly considered as a marker of a mindful and relaxed state [46], and it is, in turn, a condition that could enhance cognitive performance [47].

When the workers are compliant, HRV is also high and the activity is increased. The activity index during the scenario revealed moments where the workers had to climb the stairs, operate in uncomfortable positions, carry heavy toolboxes, etc. Notwithstanding the increase in the workload, the BR was not impacted and HR remained low when the workers were compliant with procedures. According to literature, high HRV has been correlated to a wide variety of psychophysiological states related to relaxation, emotional regulation, constructive coping with stressors, and effective attention allocation [48]. This kind of psychological state is relevant in terms of safety, since it allows the worker to effectively manage his/her cognitive and emotional resources.

### 5.2. Non Compliant Situations and Physiological Data

The pattern of physiological parameters changes in an interesting way, when the workers are non-compliant: HR increases and BR becomes negatively correlated with it. This kind of asynchrony may be correlated with lack of mindfulness and presence of stress. When the workers are non-compliant, BR is positively correlated with HRV. As workers’ BR is low and positively correlated with HRV, we can interpret this result as a sign of stress. As stated by [49], “the sudden decrease of HRV can be treated as a warning from the body systems […] and is an objective method to evaluate occupational burnout.” In addition, HR is high also when there is a decrease in the activity, which may be interpreted as a sign of anxiety and stress [49].

## 6. Conclusions

The aim of this study was to introduce high fidelity simulation as a professional training method aimed at fostering self-reflection about the technical and non-technical skills in safe operations for electricity distribution workers. In order to foster self and peer observation, we developed a tool for the observation of specific behavioral markers for electricians as single operators and as a team during simulated scenarios. In addition, we wanted to add a new sensor-based method for observing behaviors based on non-verbal cues (NVC), and to monitor physiological parameters for possible correlations with compliant behaviors.

The research has relevant implications for research, practice and society. In terms of research, it is one of the first demonstrations of a correlation between compliant behaviors and physiological parameters. This approach deserves further investigation, identifying the most reliable and sensitive parameters related to safe behaviors, in order to enrich the set of cues to be monitored for safety’s sake. In addition, we developed a set of tools (the SPS Checklist) and a framework (the simulation implementation process), which could be used in further researches in analogous fields.

Concerning the implications for practice and society, the method could be easily disseminated to other technical domains (e.g., high voltage workers, power plant workers, etc.), or even other work domains (e.g., team of fire-fighters). For the first time in the electric industry, it is possible to explicitly analyze SPS by means of a structured form, based on the specific activities. In addition, the training program has been positively evaluated by the participants and simulation could become a regular aspect of professional training for safety’s sake [2]. Moreover, the analysis of physiological patterns could be explicitly addressed during the professional training. In addition, the research is a further demonstration of the effectiveness of high-fidelity simulation as a training method, especially for competences development. The advantage of the method lies in the experiential engagement of workers followed by a structured session of debriefing; this is the core factor for raising awareness on practices and safety issues, and the advantage of using a peer-observation form like the SPS checklist is that the benefit is not only for those who took part to the simulation, but also for the observers, who learn to develop a critical perspective on their activity. This is particularly important for work activities and roles that are strongly oriented to practical operations, where the self-reflective competence may be lacking. Furthermore, high-simulation training is an effective method to be integrated with top-down safety management, since can support the training of new procedures, the discussion of “hidden practices”, the clarification of attitudes and beliefs.

Among the limitations of the present study, we list its pilot nature, since it was carried on with a limited number of participants, which does not allow us to extend the results to other populations of workers. In addition, we could not track the training effectiveness beyond the simulation session day, and further research should investigate the medium-to-long term effects of the method. In addition, the behavioral and physiological recording was analyzed only after the simulation session and by means of a time-consuming analysis of the correlations between compliance behaviors and NVC cues. This means that the NVC tracking was of little use during the debriefing session, other than a qualitative discussion of the physiological trends (e.g., a heart rate sudden increase during a critical situation) and a subjective view of what the participants were looking at (by means of the head mounted action cam).

Future developments of this research could aim at developing a new generation of sensor-based systems for monitoring team coordination in both routine and extreme situations, namely, a context simultaneously marked by high levels of uncertainty, change and risk. The analysis of social signals and face-to-face communication patterns (e.g., kinesics, proxemics, and interpersonal synchronization), could be combined with other sources of information such as survey and performance metrics (e.g., feedback on the SPS, self and peer assessment of performance efficiency, etc.). We argue that these systems could help teams to design interventions aimed at enhancing individual and group performance, especially for coordinating efficiently and ensure their resilient capacity to face risk and overcome perturbations, a critical aspect of extreme situations. A main challenge to be addressed is to continue developing the simulation-based training method adapted to these technological trends so we will guarantee that professionals could exploit these quantitative results into their practice. 

## Figures and Tables

**Figure 1 ijerph-18-01591-f001:**
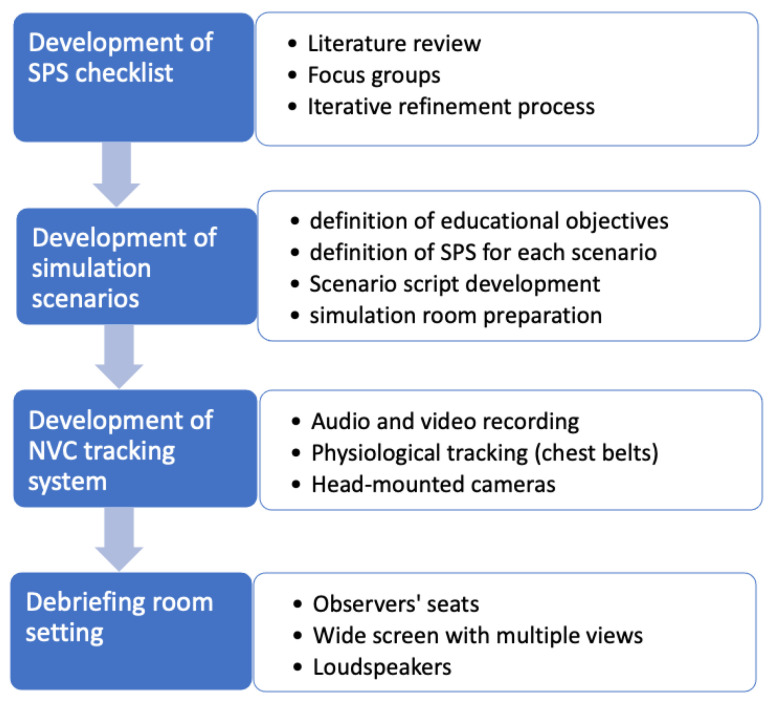
Outline of the preparation of materials.

**Figure 2 ijerph-18-01591-f002:**
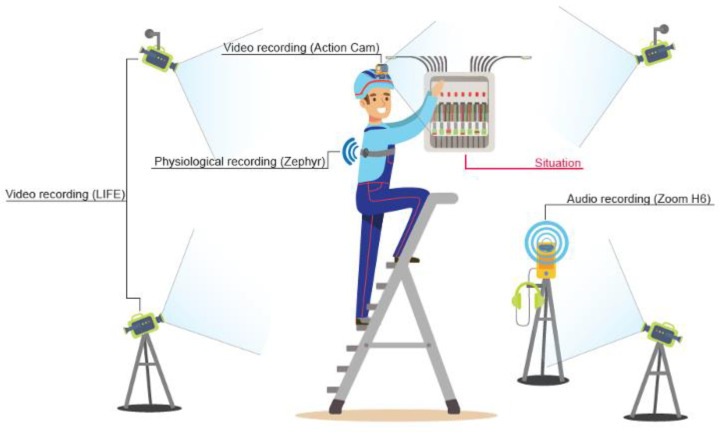
Location of the cameras.

**Figure 3 ijerph-18-01591-f003:**
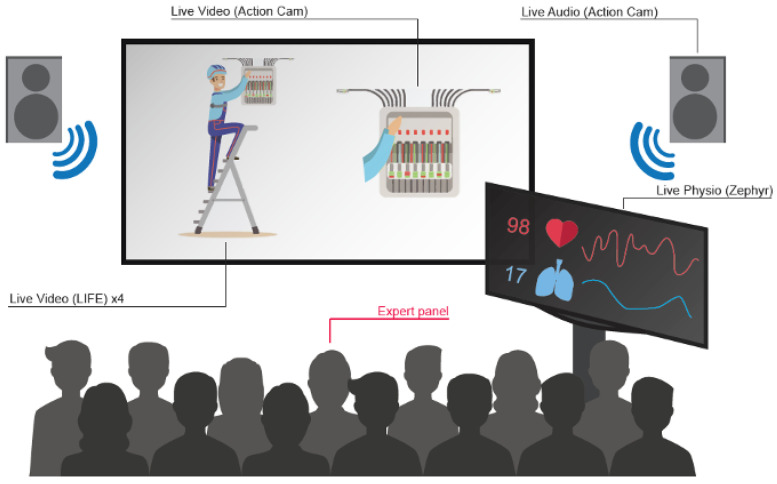
Perspective from the debriefing room.

**Figure 4 ijerph-18-01591-f004:**
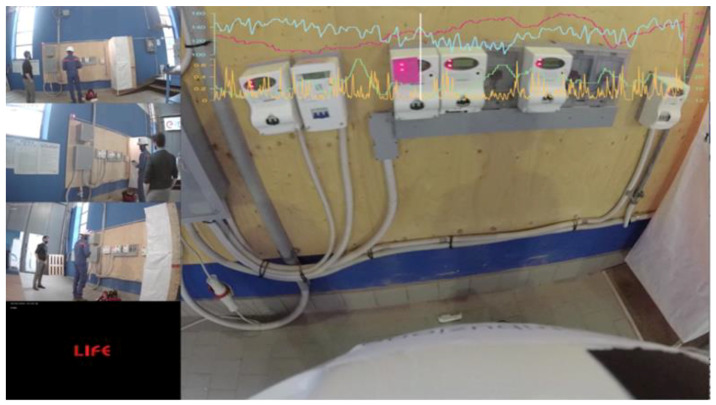
The view from the monitor in the debriefing room.

**Figure 5 ijerph-18-01591-f005:**
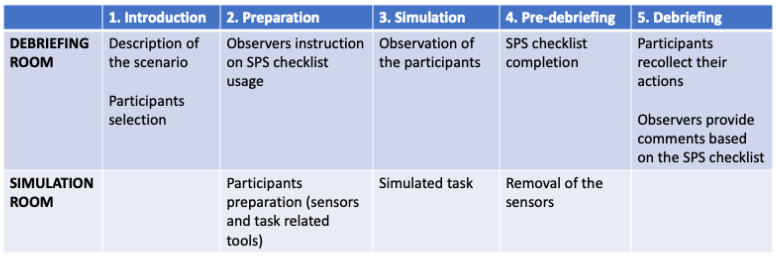
The phases of the procedure of the training.

**Figure 6 ijerph-18-01591-f006:**
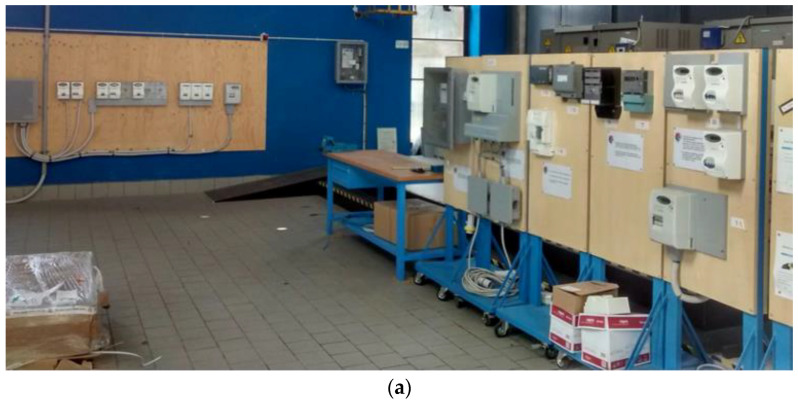
(**a**) The simulation center, and (**b**) the preparation of the scenarios

**Figure 7 ijerph-18-01591-f007:**
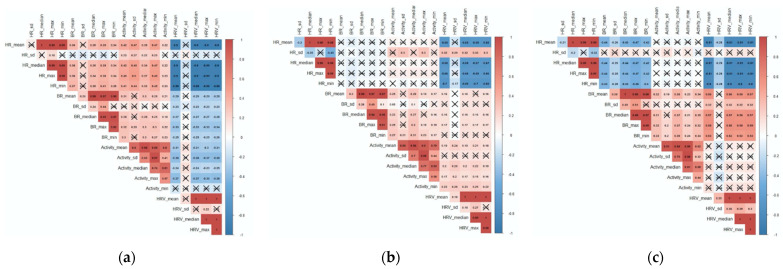
(**a**) Correlation between Non-Verbal Cues (NVC) physiological data during the compliant condition (HR = heart rate, BR = breathing rate, HRV = heart rate variability, sd = standard deviation). (**b**) Correlation between NVC physiological data during the partially compliant condition (HR = heart rate, BR = breathing rate, HRV = heart rate variability, sd = standard deviation). (**c**) Correlation between NVC physiological data during the non-compliant condition (HR = heart rate, BR = breathing rate, HRV = heart rate variability, sd = standard deviation).

**Figure 8 ijerph-18-01591-f008:**
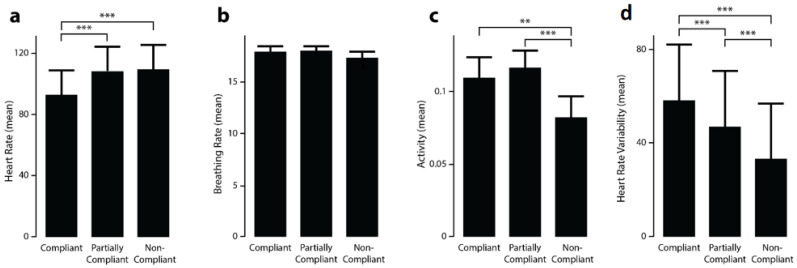
Estimate of (**a**) heart rate, (**b**) breathing rate, (**c**) activity, (**d**) heart rate variability for all 3 conditions: compliant, partially compliant, and non-compliant; ** *p* < 0.01, *** *p* < 0.001.

**Table 1 ijerph-18-01591-t001:** The matching between scenarios, the SPS and the specific behavioral markers.

SPSs	1	2	3	4	5
1. Knowledge of expected conditions	x	x	x	x	x
2. Observation of real conditions	x	x	x	x	x
3. Understanding real conditions	x	x	x	x	x
4. Implementation of safe-working conditions	x	x	x	x	x
5. Third-Party communication	x	x			x
6. Maintaining attention despite disturbances	x				
7. Team communication/collaboration		x		x	x
8. Documentation usage		x			
9. Stopping the work due to possible fraud			x		
10. Stopping the work due to unsafe conditions				x	x

**Table 2 ijerph-18-01591-t002:** Statistics of the items of the post-training feedback form.

Item	N	Mean	SD	Median	t	df	sig.	d
Non-judgmental approach	26	4.81	0.40	5	22.934	25	<0.001	4.53
Scenarios are similar to working situations	25	3.72	0.84	4	4.272	24	<0.001	0.86
Realism of scenarios	26	3.77	0.59	4	6.682	25	<0.001	1.31
Commitment	26	4.42	0.76	5	9.579	25	<0.001	1.87
Satisfaction for the training	26	4.35	0.80	5	8.611	25	<0.001	1.69
Intention to participate to other similar trainings	26	4.00	1.10	4	4.655	25	<0.001	0.91
Usefulness of the training	26	4.00	0.85	4	6.001	25	<0.001	1.18
Usefulness of the introduction	26	4.08	0.69	4	7.977	25	<0.001	1.57
Usefulness of participating in simulated activities	25	4.40	0.87	5	8.083	24	<0.001	1.61
Usefulness of watching simulated activities	26	4.35	0.75	4	9.211	25	<0.001	1.80
Usefulness of debriefing	26	4.46	0.76	5	9.799	25	<0.001	1.91

SD: Standard Deviation; df: degrees of freedom; sig.: significativity.

## Data Availability

The data presented in this study are available on request from the corresponding author. The data are not publicly available due to privacy reasons (physiological data recordings of the participants).

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
