# Peer review of "Simulation as a Training Method for Electricity Workers’ Safety"

_ijerph, 2021, doi:10.3390/ijerph18041591_

Round 1

Reviewer 1 Report

General comments

I have reviewed your manuscript and I enjoyed reading it. I will recommend that it is accepted but there are some comments you need to consider that will improve the work. I appreciate the opportunity to be among the first to read the manuscript.

Abstract

Good points are here but it is better to reduce the details on the methods and add two more key findings. There is also the need to add, at least, one recommendation.

Introduction   

Provide citations for the claim on line 43 to 45.

After very good arguments from line 26 to 84, you need to show how your research sits in the current domain of knowledge. You have already created a platform for this up from line 26 till 84, you need to conclude it there and show your contribution to the body of knowledge following on from the arguments you have already made. Then, the aims and objectives of the study can come here or earlier. The place (see line 147 to 153) that you presented the aim of the study and objective (I suppose) impacts on the readability of the work. I view that points from lines 147 to 153 should be presented earlier in the study. It took time to see what you aim to do. Recall, that the attention span of reader may be short, or they will lack the patience to keep reading. Apart from these, the rest of this section reads well.

Materials and methods

Who are the ‘other professionals’ (line 161)

More details on the focus group (FG) and the rationale for this is needed. How did you conduct the FG? Details are needed. How did you analyse the data? What are the demographics of the participants?

It will be good to underpin the entire section theoretically.

Ethical consideration and approval-related points will need to be reported.

It will be good to mention earlier in this section, all the methods adopted. I recommend a graphic presentation including the sequence of usage.

Apart from these, this section is very detailed.

Procedures

Very detailed, well done.

Results

Well written

Comments

Good points but I am missing implications for research, practice and society.

You explained the results here, well done.  

Author Response

First of all the authors want to thank Reviewer 1 for the useful comments. We followed the requests and we believed the manuscript had a significant improvement. Here we write our responses (in italics) to each of the comments by Reviewer 1.

Abstract

Good points are here but it is better to reduce the details on the methods and add two more key findings. There is also the need to add, at least, one recommendation.

Reply: we reduced the details in the methods section, and we specified the results. Namely, we added this sentence: (line 19) “An in-depth analysis of physiological indexes and behaviours compliant to safety procedures revealed that breath rate and heart rate patterns commonly related with mindful and relaxed states were correlated with compliant behaviours, and patterns typical of stress and anxiety were correlated with non-compliant behaviours.”

As a recommendation we added: (line 24) “Future research should assess the long-term effectiveness of high-fidelity simulation for electricity workers, and should investigate non-invasive and real-time methods for tracking physiological parameters.”

Introduction   

Provide citations for the claim on line 43 to 45.

Reply: we added the reference to one of the most recent and comprehensive contributions (see Ref. 5 and 6)

After very good arguments from line 26 to 84, you need to show how your research sits in the current domain of knowledge. You have already created a platform for this up from line 26 till 84, you need to conclude it there and show your contribution to the body of knowledge following on from the arguments you have already made. Then, the aims and objectives of the study can come here or earlier. The place (see line 147 to 153) that you presented the aim of the study and objective (I suppose) impacts on the readability of the work. I view that points from lines 147 to 153 should be presented earlier in the study. It took time to see what you aim to do. Recall, that the attention span of reader may be short, or they will lack the patience to keep reading. Apart from these, the rest of this section reads well.

Reply: we thank the Reviewer for this suggestion, that improves the readability of the paper. We moved the mention to the aims of the study immediately after describing the research by Saurin (line 101).

Materials and methods

Who are the ‘other professionals’ (line 161)

Reply: we specified the other professions we took into account and we added (line 200) “(e.g., healthcare practitioners)

More details on the focus group (FG) and the rationale for this is needed. How did you conduct the FG? Details are needed. How did you analyse the data? What are the demographics of the participants?

Reply: We specified that the FG was a participatory-design meeting with experts, because, as you correctly point out, the focus group is a more structured method. We added some details about participants of the interview, how we conducted the and how we used the data

(line 225) “Then, we led two participatory-design meetings with a new sample of five practitioners to validate the content of the items and to check their comprehensibility. All the participants were male expert workers (>10 years of experience) with different roles (operator, team leader and engineer). During the first meeting a facilitator went through the items one by one and asked the participant to explain them. Participants were encouraged to discuss the meaning of the items and their relevance for safety. An observer took note of the relevant information. As a result, unclear items were rephrased. Furthermore, some items were added to encompass aspects that were not considered in the first draft. The second meeting was conducted in the same way with the same participants to confirm the refined version of the instrument was clear, easy to understand, without any ambiguity or inconsistency, and complete.”

It will be good to underpin the entire section theoretically.

Reply: we added the opening sentence (line 184): “According to Thomas [11] the development of an assessment system should be based on evidence coming from some of the following sources: (i) review of the scientific literature, (ii) accident and incident analysis, (iii) cognitive task analysis, (iv) interviews/focus groups with expert operators. In our study, we mainly relied on literature review and interviews/focus groups with expert operators.”

In addition, at line 242 we provided further references concerning the guidelines for scenario development (Prince, Oser, Salas and Woodruff (1993) and, at line 263, the behavioral markers development (Klampfer, et al., 2001).

Ethical consideration and approval-related points will need to be reported.

Reply: at line 413, we reported that the participants signed an informed consent form, written according to the APA rules. As the collected questionnaire was anonymous and the research investigated psychosocial variables, not adopting a medical perspective, an ethical committee approval was not sought.

It will be good to mention earlier in this section, all the methods adopted. I recommend a graphic presentation including the sequence of usage.

Reply: we specified that the timeline of the whole procedure is available at Appendix D. In addition, we created a new graphic description both of the material preparation (fig. 1) and the procedure (fig. 5).

Apart from these, this section is very detailed.

Procedures

Very detailed, well done.

Results

Well written

Comments

Good points but I am missing implications for research, practice and society.

Reply: we specified the implications in the conclusions as follows (line 652):

“The research has relevant implications for research, practice and society. In terms of research, it is one of the first demonstrations of a correlation between compliant behaviours and physiological parameters. This approach deserves further investigation, identifying the most reliable and sensitive parameters related to safe behaviours, in order to enrich the set of cues to be monitored for safety’s sake. In addition, we developed a set of tools (the SPS Checklist) and a framework (the simulation implementation process), that could be used in further researches in analogous fields.

Concerning the implications for practice and society, the method could be easily disseminated to other technical domains (e.g., high voltage workers, power plant workers, etc.), or even other work domains (e.g., team of fire-fighters). For the first time in the electric industry, it is possible to explicitly analyse SPS by means of a structured form, based on the specific activities. In addition, the training program has been positively evaluated by the participants and simulation could become a regular aspect of professional training for safety’s sake [2]. Moreover, the analysis of physiological patterns could be explicitly addressed during the professional training. In addition, the research is a further demonstration of the effectiveness of high-fidelity simulation as a training method, especially for competences development. The advantage of the method lies in the experiential engagement of workers followed by a structured session of debriefing; this is the core factor for raising awareness on practices and safety issues, and the advantage of using a peer-observation form like the SPS checklist is that the benefit is not only for those who took part to the simulation, but also for the observers, who learn to develop a critical perspective on their activity. This is particularly important for work activities and roles that are strongly oriented to practical operations, where the self-reflective competence may be lacking. Furthermore, high-simulation training is an effective method to be integrated with top-down safety management, since can support the training of new procedures, the discussion of “hidden practices”, the clarification of attitudes and beliefs.”

Reviewer 2 Report

Simulation as a training method for electricity workers’ safety

IJERPH-1080074

Manuscript IJERPH-1080074 describes the development of a training model for electricity distribution workers, based on high fidelity simulation. Occupational safety is a relevant issue throughout the productive sectors and innovative and effective training methods are surely needed. Therefore, I appreciated the contribution given by the present work. Having said this, I think that some parts of the manuscript may benefit from further elaboration. I recommend that the article be revised and resubmitted following a thorough consideration of the attached list of comments.

General comments:

  • I think that the authors should better highlight the relevance of the study for the field of knowledge. I mean, why addressing the electrical domain? Is it because of its hazardousness? More data should be provided to support the choice of this sector: rate of accidents, injuries, effectiveness of previous training interventions and training needs.

  • The objectives of the study are not clearly stated, in my opinion. The aim is described in at least 5 different ways (see LL12-13, LL106-108, LL 132-135, LL147-153, LL527-532), from the abstract to the discussion, making it difficult for the reader to gather the specific contribution given by the study.

Abstract:

LL12-13: this statement does not appear to derive from the previous ones: what is the relationship between what is being said here and the evaluation of non-technical skills? Please clarify.

L19: the authors here introduce the construct of “self-reflection” as used it to summarize their results. Thus, I would expect this to be a key concept for the study, recurring many times throughout the manuscript. However, this term is neither clearly defined in the introduction nor it is used to interpret the obtained results: it only appears once at the end of the introduction (no clear definition given) and once in the conclusions section (where it seems to comes out of the blue). Please revise and better define this construct.

Introduction:

LL83-84: showing what? Were the virtual scenarios better than the physical ones?

LL103-105: please provide a rationale for this definition. The relationship between NTS and SPS is not immediately clear. It is not clear whether this is a definition provided by the literature or developed by the authors. In this latter case, more references to previous research are needed to clarify the basis for the development of this construct.

L114: I think the authors refer here to an observation and evaluation during the simulation? Please clarify and provide specific references to support this statement.

LL117-121: Please provide specific references to support this statement.

L126: which tools do the authors refer to? Please add references to previous studies using these tools and briefly comment on these tools, pointing out their weaknesses.

L130: Saurin and colleagues: add the proper reference number.

LL139-140: what did these studies find out? Please briefly describe these previous results.

LL144-146: again, what did this study find out? The aim was to establish the effect of simulation-induced stress upon learning, but which were the results

L151: never cited before, comes out of the blue.

Materials and methods:

Since the study included different phases, participants, and instruments, I would suggest the authors to add an “overview of the study” paragraph to describe the whole study and its steps, maybe adding also one figure to depict the entire process, for clarity of the reader.

L165: please clarify the relationship between NTS and SPS (see my previous comment on LL 103-105).

LL167: representative samples: please provide the number of participants involved. On which basis could the authors say that they were “representative of professionals”?  Please provide additional information on the sample involved and the selection procedure.

L168: “adaptation of the tools”: which tools? Provide references.

L183: new sample of practitioners: how many participants? Please clarify.

L185: please provide adequate rationale for the choice of investigating these three areas.

LL215-217: where did the definition of immersivity come from? Please provide adequate references.

L228: provide references for this definition.

L238: the reader may not be familiar with the concept of “near misses”: please define it, referring to previous literature.

Table 1: what does C stand for?

Results:

Data analysis method: please clearly separate what is ‘method’ from what is ‘results’. Move all the information regarding the data analysis to the “Methods” section (“data analysis” paragraph) and retain here only information regarding the obtained results.

L422: please provide rational for the development of the questionnaire: how was this questionnaire built? Where did the sections, scales and items used come from? Please provide additional information.

L438-9: is this classification based on previous works? If yes, please provide adequate references. If it was created by the authors, please clarify on which basis it was created.

LL452-458: the authors used a GLM to investigate causal relationships between independent and dependent variables…but based on what? How did they choose the level of compliance as the independent variable? Please refer to previous literature to support and clarify this choice.

Comments:

LL486-489: why did the authors only comment on physiological data? The aim of the study appears to be much wider. Please verify the consistency between aims and aspects considered in the discussion and revise this section accordingly.

Author Response

First of all the authors want to thank Reviewer 2 for the useful comments. We followed the requests and we believed the manuscript had a significant improvement. Here we write our responses (in italics) to each of the comments by Reviewer 2.

Abstract:

LL12-13: this statement does not appear to derive from the previous ones: what is the relationship between what is being said here and the evaluation of non-technical skills? Please clarify.

Reply: we clarified the logical link as follows (line 12):

“Therefore, a structured process to develop effective simulation scenarios and tools for the observation and feedback about performance is crucial. To this aim, in the present research, we developed…”

L19: the authors here introduce the construct of “self-reflection” as used it to summarize their results. Thus, I would expect this to be a key concept for the study, recurring many times throughout the manuscript. However, this term is neither clearly defined in the introduction nor it is used to interpret the obtained results: it only appears once at the end of the introduction (no clear definition given) and once in the conclusions section (where it seems to comes out of the blue). Please revise and better define this construct.

Reply: from line 34 we adopted a more rigorous and consistent term: reflexivity, grounding it on Schön’s theory of reflective practice. We introduced and explained this term in the introduction and we used this term consistently along the paper.

Introduction:

LL83-84: showing what? Were the virtual scenarios better than the physical ones?

Reply: It depends on the task to be simulated. At line 98, we clarified and completed the sentence, adding “… demonstrating that operational contexts based on computer interfaces could be effectively transferred into virtual simulations”

LL103-105: please provide a rationale for this definition. The relationship between NTS and SPS is not immediately clear. It is not clear whether this is a definition provided by the literature or developed by the authors. In this latter case, more references to previous research are needed to clarify the basis for the development of this construct.

Reply: from line 109, we clarified that the SPS concept is our proposal to overcome a problematic definition of SPS. We provide literature references for the criticism to NTS (Nestel et al., 2011), and for our  proposal (Brown et al., 1989).

L114: I think the authors refer here to an observation and evaluation during the simulation? Please clarify and provide specific references to support this statement.

Reply: at line 146, we clarified that the observation and evaluation is during simulations and we provided literature reference to Rudolph et al. (2001) concerning the risk of judgmental approach to the debriefing.

LL117-121: Please provide specific references to support this statement.

Reply: at line 153 we provided reference to support this statement. Specifically: Rudolph (2007) and Husebo (2013).

L126: which tools do the authors refer to? Please add references to previous studies using these tools and briefly comment on these tools, pointing out their weaknesses.

Reply: at line 156 we reported references to tools developed in three domains (healthcare, aviation, railways). The rationale of this statement is that the existing tools are specific for a given professional domain and there is lack of structured tools for electricity distribution workers.

L130: Saurin and colleagues: add the proper reference number.

Reply: we added the references at line 162

LL139-140: what did these studies find out? Please briefly describe these previous results.

Reply: we explicitly reported at line 168 that these studies found significant correlations between performance and cognitive load physiological indexes

LL144-146: again, what did this study find out? The aim was to establish the effect of simulation-induced stress upon learning, but which were the results

Reply: at line 171 we specified that the authors found that simulation-induced stress decreased when the simulation was repeated over time and learning consolidated.

L151: never cited before, comes out of the blue.

Reply: the Reviewer is right, we removed this point, because it is misleading and it is implicit in the development of the SPS training toolkit, as described later on.

Materials and methods:

Since the study included different phases, participants, and instruments, I would suggest the authors to add an “overview of the study” paragraph to describe the whole study and its steps, maybe adding also one figure to depict the entire process, for clarity of the reader.

Reply: we followed a suggestion from Reviewer 1 and we added two figures (fig 1 and fig. 5) to clarify the phases of materials preparation and the procedure

L165: please clarify the relationship between NTS and SPS (see my previous comment on LL 103-105).

Reply: we specified this relationship in the previous paragraph. Here (line 212) we clarified that we integrated “… the NTS with the technical aspects of the EDOs tasks” in order to develop a list of SPS.

LL167: representative samples: please provide the number of participants involved. On which basis could the authors say that they were “representative of professionals”? Please provide additional information on the sample involved and the selection procedure.

Reply: We specified as requested. At line 207, we wrote: “We held a series of meetings with professionals that were representative of the professional profiles involved in the research (four engineers and two operators), thereby working on the adaptation of the tools to the Italian electricity distribution context. Each meeting was held by three expert psychology researchers and involved at least one or two electricity distribution experts.”

L168: “adaptation of the tools”: which tools? Provide references.

Reply: we clarified that we adapted the tools provided in Wachs et al.

L183: new sample of practitioners: how many participants? Please clarify.

Reply: at line 225 we specified the number of participants and the rationale for this choice

L185: please provide adequate rationale for the choice of investigating these three areas.

Reply: at line 236 we specified the rationale for this choice and we wrote: “With the consensus of the experts, the items explored three main areas concerning the operators’ tasks: task management, customer management, and teamwork.”

LL215-217: where did the definition of immersivity come from? Please provide adequate references.

Reply: at line 262 we changed the sentence and explained the concept, we used the term “immersion” instead of “immersivity” and we added a literature reference

L228: provide references for this definition

Reply: at line 276 we provided the reference to McTighe & Wiggins (2004)

L238: the reader may not be familiar with the concept of “near misses”: please define it, referring to previous literature.

Reply: at line 286 we rephrased clarifying the concepts as follows: “Most of the time, scenarios are adaptation of real cases involving accidents or quasi-accidents (events when the accident did not happen but it could have easily happened).” We did not provide reference only because we did not explicitly mention the technical term “near miss” and we did not want overload the reference list

Table 1: what does C stand for?

Reply: C is the version of the scenario. We first developed two earlier versions (A and B). We think it is not a useful detail for the reader, therefore we simply renamed them from 1 to 5, removing the letter C

Results:

Data analysis method: please clearly separate what is ‘method’ from what is ‘results’. Move all the information regarding the data analysis to the “Methods” section (“data analysis” paragraph) and retain here only information regarding the obtained results.

Reply: we moved the “data analysis method” paragraph in the Methods section.

L422: please provide rational for the development of the questionnaire: how was this questionnaire built? Where did the sections, scales and items used come from? Please provide additional information.

Reply: at line 474 we specified that we adopted the Kirkpatrick’s model for the assessment of learning and we focused on the post-training reaction in terms of appreciation and perceived usefulness of the experience.

L438-9: is this classification based on previous works? If yes, please provide adequate references. If it was created by the authors, please clarify on which basis it was created.

Reply: This three-fold classification is based on the proposal of the SME who assessed the simulation performance. This approach is grounded in the electricity distribution company practice of safety inspectors to assess compliance to safety procedures of workers. They frame the compliance continuum ranging from a complete respect of safety procedures, to a partial respect, where some aspects are missing or some steps pf a procedure are swapped, to a clear violation, where some critical aspects are missing of the entire procedure is violated. This three-fold classification allows the inspectors a sufficient ease of use, with enough granularity for the adequate assessment of the behaviour.

Therefore, at line 492 we wrote as follows: “In order to have a detailed analysis of the behaviour of each participant to the simulation, we performed a further assessment with the support of an expert of the electrical operations, provided by the electricity distribution company. The assessment consisted in the analysis of each video recording, tracking each behaviour and classifying it.The classification was based on the procedures of the company known by the expert. The expert evaluated if they were completely respected (compliant), if some minor issue were detected the behavior was labeled as partially-compliant, or if there were major violations it was labeled as non-compliant.”

LL452-458: the authors used a GLM to investigate causal relationships between independent and dependent variables...but based on what? How did they choose the level of compliance as the independent variable? Please refer to previous literature to support and clarify this choice.

Reply: We emphasized in the paper that the three levels of compliance to safety procedures (compliant, partially compliant and non-compliant) were evaluated by subject matter experts (SME) through video analysis based on field expertise and years of practice. From line 525 we also provided more explanations as to why we use correlation and linear mixed models.

Comments:

LL486-489: why did the authors only comment on physiological data? The aim of the study appears to be much wider. Please verify the consistency between aims and aspects considered in the discussion and revise this section accordingly.

Reply: From line 572 we provided an extensive comment on the results of the post-training questionnaire, linking the results to the aims of the research and the most relevant issues concerning simulation-based training.

We also rewrote the comments on the physiological data (line 600 onwards), in order to highlight the most relevant results.

Round 2

Reviewer 2 Report

The authors did a very good job in addressing all my previous concerns.

Just two final minor comments:

L210: please remove the term "representative",  since from a statistical point of view it has a very specific meaning and it is related to specific sampling methods, which were not adopted for the present study. You could change it into "professionals belonging to/from different professional profiles..." or something similar.

L252: please check the consistency in the use of "immersion", since "immersivity" still occurs 4 times throughout the text.

Author Response

The authors did a very good job in addressing all my previous concerns.

Reply: the authors thank Reviewer 2 for the insightful comments, which helped to improve the quality of the manuscript

L210: please remove the term "representative",  since from a statistical point of view it has a very specific meaning and it is related to specific sampling methods, which were not adopted for the present study. You could change it into "professionals belonging to/from different professional profiles..." or something similar.

Reply: at line 207 we changed as suggested: "with professionals that were belonging to different professional profiles". 

L252: please check the consistency in the use of "immersion", since "immersivity" still occurs 4 times throughout the text.

Reply: we checked the manuscript and we changed "immersivity" with "immersion", as suggested